# Induced Topological Superconductivity in a BiSbTeSe_2_-Based Josephson Junction

**DOI:** 10.3390/nano10040794

**Published:** 2020-04-21

**Authors:** Bob de Ronde, Chuan Li, Yingkai Huang, Alexander Brinkman

**Affiliations:** 1MESA+ Institute for Nanotechnology, University of Twente, 7522 NB Enschede, The Netherlands; bobdronde@gmail.com (B.d.R.); chuan.li@utwente.nl (C.L.); 2Van der Waals–Zeeman Institute, IoP, University of Amsterdam, 1098 XH Amsterdam, The Netherlands; y.huang@uva.nl

**Keywords:** Topological insulator, Josephson junction, Majorana mode

## Abstract

A 4π-periodic supercurrent through a Josephson junction can be a consequence of the presence of Majorana bound states. A systematic study of the radio frequency response for several temperatures and frequencies yields a concrete protocol for examining the 4π-periodic contribution to the supercurrent. This work also reports the observation of a 4π-periodic contribution to the supercurrent in BiSbTeSe${}_{2}$-based Josephson junctions. As a response to irradiation by radio frequency waves, the junctions showed an absence of the first Shapiro step. At high irradiation power, a qualitative correspondence to a model including a 4π-periodic component to the supercurrent is found.

Topological insulators have been a popular topic of research for over a decade now. With potential applications from spintronics [1,2] to topological quantum computation [3,4], especially the transport properties are heavily studied in topological insulators. Specifically, the interface between topological insulators and superconductors is of interest, the vicinity of the latter being able to induce topological superconductivity in the former. Topological superconductivity makes a compelling objective since it is predicted to host Majorana states, which can act as the basic elements needed to perform topological quantum computation. Signatures of Majorana states have been found in several systems, such as nanowires [5,6,7,8], atomic chains [9] and Dirac semimetals [10,11]. Signatures have also been found in a proximitized Bi${}_{2}$Te${}_{3}$ thin layer by STM measurement [12], and in Bi${}_{2}$Se${}_{3}$-based [13], (Bi${}_{0.06}$Sb${}_{0.94}$)${}_{2}$Te${}_{3}$ [14] and HgTe-based Josephson junctions [15,16].

One of the used techniques to probe the topological features of a Josephson junction is to measure 4π-periodic contributions to the current–phase relation. The 4π-periodicity arises from a topologically forbidden back-scattering of quasiparticles in an Andreev bound state. However, since quasiparticle poisoning can destroy this state, one has to measure fast enough and that is why the effect cannot be measured in direct-current experiments and why radio-frequency experiments are performed. A 4π-periodic current–phase relation would lead to the absence of odd-numbered Shapiro steps in the current–voltage characteristic of a junction under microwave irradiation. A missing n=1 Shapiro step was first reported [15] for strained HgTe, a 3D topological insulator with 2D topological surface states. The presence of the higher order odd Shapiro-steps (e.g., the third and fifth steps) can either be explained by capacitance in the circuit [17] or by Joule heating in the junction [13]. The frequency dependence of the effect can be used to distinguish [15] the topological bound state observation from more mundane explanations of the effect such as Landau–Zener tunneling between bound states which is more likely at higher frequencies. For a HgTe quantum well, forming a 2D topological insulator with 1D topological edge states, also the higher order odd steps were shown to be absent [16]. The 1D edge states have the advantage that scattering is always forbidden, whereas in two dimensions bound states occur in non-perpendicular directions that are 2π-periodic [3,18]. Later, similar experiments revealed missing Shapiro steps for a variety of topological Josephson junctions [10,11,13,14] as well as halved Josephson radiation frequencies of voltage biased topological Josephson junctions [19].

Here, the measurement of Majorana signatures in the three-dimensional topological insulator BiSbTeSe${}_{2}$ is reported. The material BiSbTeSe${}_{2}$ [20,21] combines the scalability of a two-dimensional transport environment with gate tunability, a combination which shows great promise for applications. The Bi-based topological insulators have the added advantage of having a larger bulk band gap when compared to (strained) HgTe, and the stoichiometric compounds have been shown to be gate tunable into the regime of negligible bulk conduction [21]. After describing the device fabrication of the BiSbTeSe${}_{2}$-based Josephson junctions and some basic material characterization, this work unveils a 4π-periodic contribution to the supercurrent through radio frequency measurements. Some challenges in distilling 4π periodicity from the radio frequency response and how to overcome them are also highlighted.

High quality BiSbTeSe${}_{2}$ single crystals were grown using a modified Bridgman method. Stoichiometric amounts of the high purity elements Bi (99.999%), Sb (99.9999%), Te (99.9999%) and Se (99.9995%) were sealed in an evacuated quartz tube and placed vertically in a tube furnace. The material was kept at 850 °C for three days and then cooled down to 500 °C at a speed of 3 °C per hour, followed by cooling to room temperature at a speed of 10 °C per minute.

BiSbTeSe${}_{2}$ crystal flakes, were mechanically exfoliated onto a p-doped Si substrate capped with a SiO${}_{2}$ layer. Measurement contacts with a width of 500 nm were patterned onto selected flakes using electron-beam lithography with a standard dose. The exposed area was etched to avoid a big height difference between the contacts and the flake. Next, parallel Nb contacts were sputter deposited in situ, capped with a thin Pd layer, and finalized by lift-off. The Nb contacts were spaced between 200 and 250 nm apart to allow the induction of a supercurrent through the BiSbTeSe${}_{2}$.

An h-BN flake was placed on top of the BiSbTeSe${}_{2}$ flake to serve as a top gate dielectric and a protection layer. We first prepared the PDMS/PMMA (1mm/A4-500 rpm) double layer on a piece of transparent glass, followed by a 10 min baking process at 70 °C. Then, the h-BN flakes were directly exfoliated onto the double layer. A suitable h-BN flake was identified under an optical microscope, after which it was put on top of the desired BSTS (1112) flake using a micro-manipulator. At the end, the entire PMMA layer was released by heating up the sample up to 130 °C. The Au top gate contact was sputter deposited onto the structure at high gas pressure and low bias voltage to avoid leakage through the h-BN. A schematic image of a typical sample is shown in Figure 1a. The fabrication of the 200 nm long junction of device 2 is the same as that of device 1, except for the ALD growth of an Al${}_{2}$O${}_{3}$ gate dielectric at 100 °C instead of the h-BN flake.

The BiSbTeSe${}_{2}$ Josephson junctions were cooled down to around 20 mK and measured inside an Oxford Instruments Triton dilution fridge. The junctions were characterized in a pseudo four-wire configuration with a combination of ac and dc current biasing. We show the result of two different devices: device 1 with junctions 1a and 1b, device 2 with one junction. An IV-curve of junction 1a at 1 K is shown in Figure 1b. The 200 nm long junction has a critical current, $I_{c}$, of 1.2 μA and an $I_{c}R_{N}$ ($R_{N}$ is the normal state resistance) product of 65 μV at low temperature. The IV-characteristic is symmetric at 1 K, in contrast to the re-trapping current observed at low temperature. This feature will help with the interpretation of the radio frequency response below.

The graph in Figure 1c shows the temperature dependence of the $I_{c}$ of junction 1a. The measured data could be fitted with a model by Galaktionov and Zaikin [22], based on the Eilenberger equations. A critical temperature, $T_{c}$, of 2 K was used in the modeling, corresponding to an induced gap, Δ, of 0.3 meV. The fit yielded an interface transparancy, *D*, of 0.78, comparable to the results of Veldhorst et al. [23], and a Fermi velocity, $v_{F}$, of 3.1·10${}^{5}$ m/s, close to what is expected [24]. From these values an estimate for the coherence length, $\xi =\frac{\hbar v_{F}}{\pi \Delta }$, of 214 nm could be calculated.

The Josephson devices are in the short junction regime, where the length (in our convention, the width *W*) is shorter than the Josephson penetration depth, $\lambda_{J}=\sqrt{\frac{\Phi_{0}}{2\pi LI_{c}}}$. At low temperatures, the inductance *L* is dominated by the kinetic inductance ($L_{K}$). We estimate the $L_{K}$ in our experiment as $L_{K}=\frac{m}{n_{s}e^{2}}l/A$ (with *m* being the mass and $n_{s}$ the superfluid density) which is about 1 pH. With a critical current of about 1 μA, the small junction regime is achieved.

In response to an applied perpendicular magnetic field, *B*, the $I_{c}$ displays a Fraunhofer pattern, as shown in Figure 1d. The period of the oscillation in *B* for the first few lobes is 6.3 G. This corresponds to an effective area of $\frac{h}{2e\Delta B}=3.3$μm${}^{2}$, where ΔB is the oscillation period. For a junction width of 4.2 μm, this leads to an effective length of 780 nm. This value is reasonable considering the flux focusing effect [25]. As can be seen in the inset of Figure 1d, the Fourier transform of the Fraunhofer pattern indicates a fairly homogeneous current flow through the crystal flake.

The uniform current distribution shows the quality of the crystal in this device. From gating measurements on a different device, a mobility, μ=Δσd/(ΔVϵ), of ∼0.3 m${}^{2}V^{-1}s^{-1}$ was obtained. This yields a mean free path of $\ell =\frac{m^{*}v_{F}\mu }{e}\approx 159$ nm, using an effective electron mass of $m^{*}=0.3m_{e}$ [24]. This is lower than the requirement for ballistic transport through the 200 nm junction of device 1. However, it is close enough to allow some ballistic channels to form, taking into account that the effective junction length might be reduced by fabrication inaccuracies. The crystal being in this intermediate regime between diffusive and ballistic transport is also reflected in the sensitivity of the system to surface mobility and fabrication treatment. Producing a device of sufficient quality to show 4π periodicity has proven challenging. Only the two junctions with a h-BN protection layer reveal the 4π periodicity, whereas more than 100 junctions fabricated without h-BN never showed this effect. The addition of the h-BN protection layer to the latest devices most likely prevents the BiSbTeSe${}_{2}$ surface from degrading to the diffusive regime, as has been seen before in BiSbTeSe${}_{2}$-based systems [26,27].

A 4π periodicity in the radio frequency response of Josephson junctions of topological materials is an indication of the presence of a zero energy Andreev bound state [10,11,15]. This phenomenon is strongest at a low excitation frequency and vanishes at higher frequency, which can be explained in a resistively shunted junction model on the basis of a competition between the visibility of the 4π-periodic and 2π-periodic states [15]. The frequency dependence of the radio frequency response was therefore studied in device 1, as shown in Figure 2. Figure 2a shows an IV-curve at medium radio frequency excitation power in a regime where Shapiro steps are visible, but superconductivity has not been fully suppressed yet. The small critical current makes noise an important source of rounding of the Shapiro steps. These curves are quantified by counting the amount of data-points in a set of voltage bins, as shown in the right hand plot of Figure 2a.

This binning data were collected for all excitation powers and is presented graphically in the form of a color map. Figure 2b shows such a binning map of junction 1a at a frequency of 4.7 GHz. The color scale is adjusted to clearly show the features of the map, gray indicates a bin count value above the color scale. At integer values of the Shapiro step size voltage at 4.7 GHz, the higher bin count indicates the presence of Shapiro steps. All integer values are represented in this measurement, except for the first. This missing first step is an important indication of the presence of a 4π-periodic component to the supercurrent [10,11,15]. Traces of the integer Shapiro step voltage values clarify the absence of the first step until the radio frequency irradiation becomes too strong for the junction to sustain a supercurrent, as can be seen in Figure 2c. In contrast to the data in Figure 2b, the binning map at a frequency of 9.6 GHz in Figure 2d clearly shows the presence of all integer Shapiro steps, including the first. Figure 2e presents the traces at integer Shapiro step size values of this binning map, clearly showing all integer Shapiro steps.

The frequency dependence of the missing first step was further investigated by defining the ratio between the first and the second Shapiro step, $Q_{12}$, as the quotient of the first maxima of the traces at hf/2e and 2hf/2e. The quotient $Q_{12}$ was calculated for a range of irradiation frequencies on junction 1a, and is presented in Figure 2f. After a certain threshold frequency, the curve clearly shows a rapid increase of $Q_{12}$, by virtue of an increase of the step height of the first step. This frequency dependence excludes conventional Landau–Zener tunneling as the cause of the 4π periodicity [28].

At low temperature, the re-trapping voltage might obscure low order Shapiro steps at low frequency, since the Shapiro step voltage scales with voltage and frequency. This possible explanation of the data were looked into by increasing the temperature to reduce the re-trapping current. The exact threshold where a Shapiro step should be visible can be difficult to determine. However, when the gap edges of an IV-trace are no longer sharp, all Shapiro steps are expected to be present. The effect of an increase in temperature on the radio frequency response was studied in device 1. This led to the conclusion that the first Shapiro step is disproportionately suppressed in the BiSbTeSe${}_{2}$ junction even when taking the low temperature opacity into account. The results of these measurements on junctions 1a and 1b, and their analysis, are included in the Appendix A.

For further comparison between the measurements and the expectation for a 4π-periodic contribution, the step size of the zeroth order Shapiro step was modeled. The resistivelyshunted junction model was used to calculate the junction voltage in the presence of radio frequency irradiation. Using the equation $i_{dc}+i_{rf}\sin\left(\Omega \tau \right)=i_{c}^{2\pi }\sin\phi +i_{c}^{4\pi }\sin\frac{\phi }{2}+\frac{d\phi }{d\tau }$, where $\tau =tR_{N}\frac{2e}{\hbar }$, $i=\frac{I}{I_{c}}$, $\Omega =\frac{f}{f_{c}}$ with $f_{c}=I_{c}R_{N}\frac{2e}{\hbar }$, and $i_{c}^{2\pi }$ and $i_{c}^{4\pi }$ are the 2π- and 4π-periodic contributions to the supercurrent, respectively [11,29]. Figure 3a shows the modeled step size without a 4π-periodic contribution and in Figure 3b a 5% 4π-periodic component was added to the supercurrent. A qualitative comparison with the measured data at 4.7 GHz in junction 1a, as shown in Figure 3c, leads to the conclusion that the inclusion of a small 4π-periodic component resembles the data much better. Without the presence of any 4π periodicity, the step size reduces to zero at each minimum of the Bessel function, while the first and third minima retain a finite value upon introduction of a 5% 4π-periodic component to the supercurrent. These finite minima are also observed, suggesting the presence of 4π-periodic modes in the Josephson junction. It is interesting to note that the lack of suppression holds for the first minimum (and subsequent odd minima) in the critical current versus applied microwave power, whereas the second minimum is very close to zero, just like the RSJ model predicts.

Control over the Fermi level can be a useful tool in measuring Dirac cone physics in topological insulators, as shown by Ghatak et al. [26]. Unfortunately, the supercurrent of device 1 could not be tuned by means of gating, presumably due to the gap between the h-BN gate dielectric and the BiSbTeSe${}_{2}$ flake being too large. However, the graphs in Figure 4 show top gate control of the supercurrent in device 2, a device for which the 200 nm long junction has an atomic layer deposited Al${}_{2}$O${}_{3}$ gate dielectric instead of the h-BN flake. Figure 4a shows the change of $I_{c}$ and $R_{N}$ as a function of gate voltage. For negative gate voltages, the critical current is almost tuned to zero due to the low charge carrier density close to the surface state Dirac point [27,30]. In panel Figure 4b, the $I_{c}R_{N}$ product shows that the induced gap is also tuned down close to the Dirac point. The proximity of the Dirac point was further investigated by examining the gate dependence of the Fraunhofer pattern, elaborated upon in the Appendix A. This data suggest that a gate could be used to isolate bound states close to the Dirac point in this type of sample.

The protection of the top surface of a BiSbTeSe${}_{2}$-based Josephson junction by a h-BN flake avoids degradation, thereby allowing for ballistic transport across the Josephson junction unveiling the presence of a 4π-periodic Andreev bound state contribution to the supercurrent. A careful study of the radio frequency response as a function of temperature and frequency is used as a protocol for examining the 4π periodicity. The indication of the presence of Majorana bound states in the system show its promise as a platform to configure a topological quantum computer. The natural scalability of the three-dimensional topological insulator (i.e., Josephson junctions can be arranged in circuits on a three-dimensional topological insulator surface [3]) and its gate-tunability only strengthen its potential for applications.

## Figures and Tables

**Figure 1 nanomaterials-10-00794-f001:**
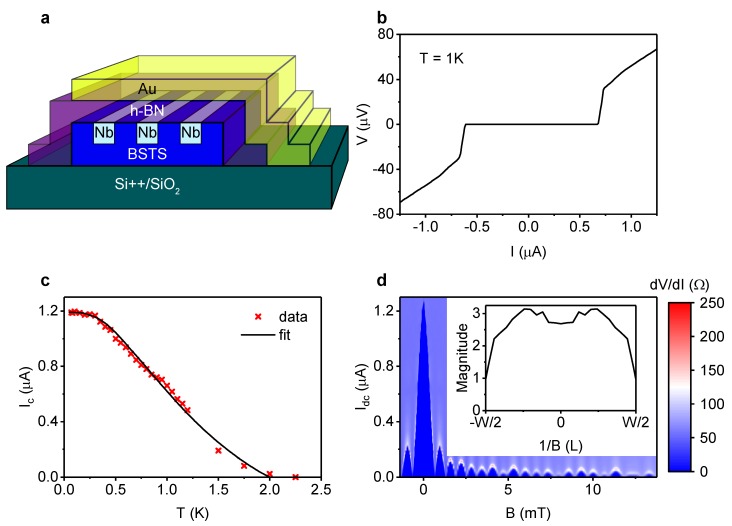
(**a**) Schematic image of a typical device. A BiSbTeSe${}_{2}$ flake was contacted with parallel Nb leads designed to create Josephson junctions in the BiSbTeSe${}_{2}$. The leads were equipped with contact pads at both ends to allow for pseudo four-wire measurements. An h-BN flake and a Au contact on top of the BiSbTeSe${}_{2}$ flake form a top gate structure. (**b**) IV-curve of junction 1a, one of the junctions of device 1, at 1 K. The critical current and the re-trapping current are similar. (**c**) Critical current as function of temperature for junction 1a. The fit with the clean limit Eilenberger equations suggests that the junction is in the ballistic regime. (**d**) Fraunhofer pattern of junction 1a. The supercurrent maximum at non-zero field indicates the presence of a small stray field. The Fourier transform in the inset shows a moderately homogeneous current distribution through the junction, except for a reduction at the edges of the flake.

**Figure 2 nanomaterials-10-00794-f002:**
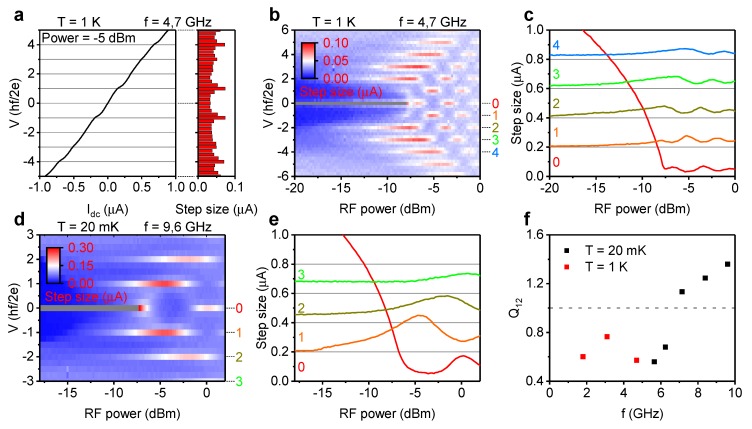
Radio frequency response of junction 1a. All data in this figure were obtained by sweeping from negative to positive bias voltage. (**a**) IV-curve under radio frequency irradiation of junction 1a in the left hand graph. The right side graph shows the voltage binning data from the IV-curve on the left. (**b**) Voltage binning data for a range of irradiation powers at 4.7 GHz of junction 1a at 1 K. The first Shapiro step is clearly absent. The scale has been adjusted to clarify the features; gray indicates the data-point falls above the scale. (**c**) Integer Shapiro step size for a range of radio frequency powers at 4.7 GHz of junction 1a. Subsequent curves have been shifted by 0.2 μA for clarity. (**d**) Voltage binning data for a range of radio frequency powers at 9.6 GHz of junction 1a. The first Shapiro step is clearly present. (**e**) Integer Shapiro step size for a range of radio frequency powers at 9.6 GHz of junction 1a. (**f**) The quotient of the first and second Shapiro steps, $Q_{12}$, of junction 1a for several frequencies.

**Figure 3 nanomaterials-10-00794-f003:**
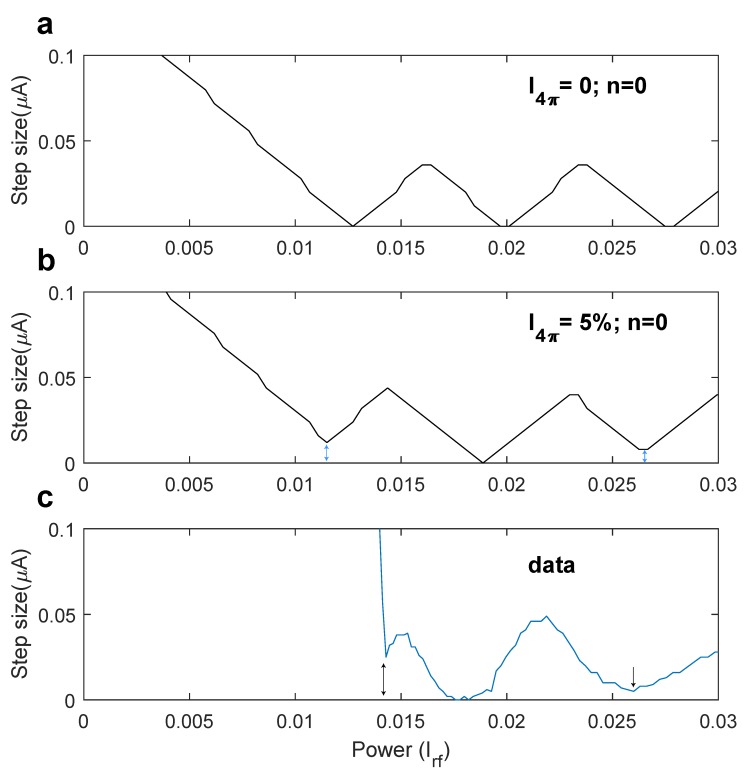
(**a**) Modeled step size of the zeroth order Shapiro step. No 4π-periodic contribution to the supercurrent was added. The step size reduces to zero periodically. (**b**) Modeled step size of the zeroth order Shapiro step with a 5% 4π-periodic contribution to the supercurrent. The step size does not completely reduce to zero at odd minima in the function. (**c**) Extracted step size data of junction 1a at 20 mK and 4.7 GHz radio frequency irradiation. The second minimum dips down deeper than the first and third minima, suggesting the presence of a 4π-periodic component of the supercurrent.

**Figure 4 nanomaterials-10-00794-f004:**
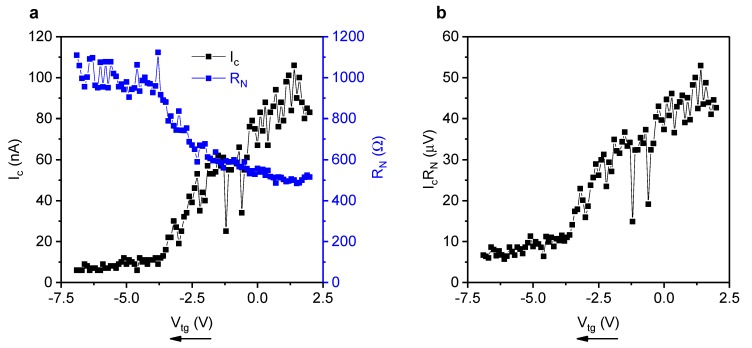
(**a**) $I_{c}$ and $R_{N}$ of device 2 for a range of top gate voltages. The supercurrent can be tuned close to zero at negative gate voltage. The arrow indicates the sweep direction of the gate voltage. (**b**) $I_{c}R_{N}$ product of device 2 for a range of top gate voltages. The induced gap could be tuned by almost an order of magnitude over the voltage range.

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
