# Peer review of "Induced Topological Superconductivity in a BiSbTeSe2-Based Josephson Junction"

_nanomaterials, 2020, doi:10.3390/nano10040794_

Round 1

Reviewer 1 Report

The manuscript reports the experimental transport characterization of the BiSbTeSe2-based Josephson junction.  In particular, the measurements of the radio frequency irradiation response of the device that is based on topological insulator reveal a 4pi-periodic contribution to the supercurrent that is suggested to be a signature of Majorana bound states. The results are important from the point of view  of application in spintronics.

The experimental work is carefully performed and the discussion well written. I recommend the manuscript for publication in Nanomaterials after some minor revisions.

  1. Lines 21-23. Some references concerning the title material are needed here.
  2. The Authors mention that only two out of the 100 produced devices display the 4pi periodicity, and only results obtained for these two are included in the manuscript. Nevertheless I was a little bit confused when reading about “junction 1a”, a “different device” (Line 67) or “junction 2” mentioned in the manuscript. For example, what is the reason for the signature “1a” instead of 1? I recommend to add a clear description of the investigated devices in the experimental part and to take care when referring to them in the discussion. Please move the sentence about the fabrication of the device 2 to the experimental part.

Author Response

We thank the Referee for the positive judgement of our manuscript and for the suggestions to further improve the manuscript. We have taken the suggestions into account by:

  1. We have added references for the used material BSTS.
  2. We have moved the sentence on the fabrication of device 2 to the experimental part. Device 1 and 2 result from two different fabrication runs. Each device had multiple junctions. Of device 1 we show junctions 1a and 1b. We have clarified this in the text and consistently refer to junction 1a instead of device 1a, etc. Also we have improved the sentence on the 100 samples. Namely, when we make a device with the hBN top gate, then we observe the 4pi periodicity (100% success rate), whereas the >100 samples made in other ways did not reveal this (0% success).

Reviewer 2 Report

The paper is aimed to investigate induced topological superconductivity in a BiSbTeSe Josephson junctions and observe 4pi-periodic supercurrent. While the paper is interesting and sound, the presented results are not fully convincing. I suggest the authors to make some clarifications and redraw the figures, after which the manuscript can be further considered for publication.

Comments:

  1. I suggest enlarging the main figures to improve their visibility. This concerns, first of all, Fig. 1b (where I suggest to add few more curves for lower temperatures), Fig. 1d, Fig. 2a.
  2. The Fraunhofer pattern in Fig. 2d as blue on light blue color is hardly visible, I suggest changing the color scale and enlarge the figure by at least a factor of 2.
  3. In Fig. 2a the Shapiro steps are actually absent, I suggest to redraw the figure for another ac power, e.g. for -5 dBm, where Shapiro steps must be enhanced. I also suggest adding the inset with few enlarged Shapiro steps, so their tilt would be more visible. Since the authors use Triton cryostat, which I assume is dry and operates with pumps, some acoustic noise can transfer to electric low frequency interference which can smooth the steps.
  4. In the caption of Fig. 2 it is written:

"figure was obtained by sweeping from negative to positive bias voltage" - does it mean that the measurements were performed in voltage bias regime? Then, it should be written explicitly in the text, since Josephson junctions are usually measured in current bias regime.

  1. In the main text it is written that:

"In contrast to the data in Figure 2b, the binning map at a frequency of 9.6 GHz in Figure 2d clearly shows the presence of all integer Shapiro steps, including the first."

However, in the caption of Fig. 2 it is written that for both driving frequencies 4.7 and 9.6 GHz, so in Fig. 2b and Fig. 2d: "The first Shapiro step is clearly absent." Please, clarify this point.

  1. My main concern is the following: the authors claim that since the Fraunhofer pattern does not approach zero, it can be an indication that there is a 4pi periodicity, which is not convincing. First, the authors perform measurements in a large temperature interval, which is nice. But since the system is rather new, not falling to zero can not be a proof. For example, for high temperature superconductors the "Fraunhofer patterns" are quite diverse, and can be not Fraunhofer at all, without approaching zero, see, e.g., Fig. 37 in Rev. Mod. Phys., 74, 485 (2002). And the particular shape of "Fraunhofer patterns" can strongly depend on temperature, see, e.g., IEEE Trans. Appl. Supercond., 28, 1100505 (2018). So, a number of additional arguments must be given to convince the reader about 4pi periodicity.
  2. The considered model is rather simple, and deals with a short overdamped Josephson junction. The authors should at least give arguments that even at low temperature of 20 mK their junction is still small, well below the Josephson length, so one can use their model and should not recourse to sine-Gordon equation. If the junction is long, one should consider more complex model and estimate possible external magnetic field, since at a certain parameters one can observe complex quasi-chaotic dynamics of Josephson vortices which can give rise of both fractional Shapiro steps and lead to suppression of integer steps, see again IEEE Trans. Appl. Supercond., 28, 1100505 (2018).

Author Response

1-3. Figure sizes and color schemes can, of course, be adjusted after consultation and upon request of the editor. It is of course a matter of taste but we are of the opinion that the preprint representation (figures with full page width) makes the figures large enough. Fig. 2a, in fact, does show the Shapiro steps. The fact that they are rounded is, in fact, a topic of the paper.

  1. The measurement are done by standard current biasing, from negative to positive. The figure only presents the measured voltage. We mentioned the measurement configuration and we now write explicitly: “The junctions were characterized in a pseudo four-wire configuration with a combination of ac and dc current biasing.”
  2. We thank the Referee for the attentive reading. Indeed, this is a misprint, see also our reply to Referee 1. The caption of fig. 2d has been corrected to say, of course, that the first Shapiro step is present.
  3. We consistently use the fact that the power dependence of the step does not go to zero, not the Fraunhofer magnetic field dependence which the Referee refers to.
  4. The sine-Gordon equation only becomes necessary when the length (in our convention, the width W) is larger than the Josephson penetration length, $\lambda_J=\sqrt{\Phi_0/2\pi L I_c}$. Usually it is used in junctions with mA range critical current. At low temperatures, the inductance L is dominated by the kinetic inductance (Lk) in superconducting devices. We estimate the Lk in our experiment as Lk = (m/2 n_s e^2) l/A which is about 1 pH, and the critical current I_c is about 1 microAmp. This would safely put us in the short junction regime. The paper that referee suggested does not not have any power dependence of the Shapiro steps, which makes it hard to compare to (only the non-oscillatory regime is relevant for the missing steps).

Reviewer 3 Report

In their paper, de Ronde et al discuss signatures of topological superconductivity in a BiSeTeSe2 sample. The paper starts with a detailed report on the fabrication and characterisation of the sample, from which it is clear that superconductivity is well-established in the junction. This part of the manuscript is very nicely readable and strikes me as a thorough experimental report, which I would like to highlight. The main achievement of the paper is presented in its second part, namely the observation of signatures consistent with topological superconductivity. More precisely, the authors study the radio frequency response of their Josephson junction. They clearly show that the first Shapiro step disappears below a certain threshold frequency, which is known to be a fingerprint of Majorana bound states, and hence of topological superconductivity. The authors also study a shunted junction model with 2pi and 4pi-periodic contributions to the supercurrent as a model for the system, and find decent agreement between theory and experiment. I therefore agree with the authors that the measured data is consistent with signatures of topological superconductivity.

While individual signatures of topological superconductivity are by now not a rare sight anymore, the field is in need for more than individual pieces of the puzzle. It seems to me that the physics of topological bound states is - fortunately, from a perspective of science - more intricate than many hoped in the beginning. This paper in my view can help solidifying the state of the field if it combines its observation with an enlarged introduction that further summarizes what signatures of Majorana bound states in RF-Josephson junction have already been seen, what signatures are missing, and what competing explanations are (quite a bit of work has been done in the past on that). Also, the paper would benefit from further elaborating on where the current results stand in that context - which observations are truly new, and which observations are missing from the current data to more reliably conclude that there really are Majorana bound states (from a simplistic theory perspective, one would for example expect also the 3rd, 5th, .. Shapiro steps to be suppressed at low frequencies, just to name one point that could be discussed)?

As I wrote above, the paper is in my view a careful analysis of a nice sample with encouraging results - and as I wrote, it is my belief that the field needs careful and honest analyses of data to point out where we need to go from here. I therefore recommend publication when the authors have addressed my above concerns.

Author Response

We thank the Referee for the positive judgement of our manuscript and for the suggestions to enlarge the introduction and discussion. We have taken the advice and the new paragraphs in the introduction are indicated in blue in the revised version of the manuscript.

Specifically, in the introduction we have added a paragraph, summarizing the 4pi-bound state observations in RF experiments and more lines in the introduction on what makes it interesting to study these in the material BiSbTeSe2. Also the presence of the higher order odd steps are discussed as well as alternative mechanisms and how to distinguish the two. Especially the alternative explanation of a retrapping current masking the first step is a topic of investigation in this manuscript.

Reviewer 4 Report

The manuscript reports measurements of the Josephson effect in BiSbTeSe_2-based superconducting hybrid systems. The authors performed magnetic field and microwave irradiation experiments.  BiSbTeSe_2 has Dirac surface (it is a topological insulator) states that can carry a supercurrent in such a Josephson junction. The Dirac nature of spin-momentum locked surface states induce Andreev bound states in a Josephson junction which are gapless and, as a consequence, the current phase relation is predicted to be 4Pi-periodic due to this mode. These zero-energy states are Majorana zero modes (MZMs) that are currently investigated in the solid state community because they might be used for a qubit. The detailed Shapiro step measurements hint on the existence of such a 4Pi Josephson effect allowing for the existence of MZMs. The main reported effect is the absence of the first (odd) Shapiro step at small frequencies and weak rf-power.

I should say here, that I could not see any supplementary material with this paper, although the authors refer to it more than once in the paper. So, my review only concerns the main text with the figures.

The experiments are well presented and are convincing since many features of the Shapiro steps as step widths, power and frequency dependences are shown. Also, the samples have been investigated according to their purities (e.g. via Fraunhofer pattern inspections). Some features could be modeled by an RSJ model analysis (suggesting that only a small fraction of the Josephson current has a 4Pi periodicity.)

Besides the convincing results and the nice presentation, the discussion of the implications of this work could be improved. The following questions and suggestions should be considered by the authors:

  1. The results resemble in many aspects similar investigations in other material systems like the surface states of strained HgTe reported in the cited Reference [15] by the Würzburg group (Wiedenmann et al., Nat. Comm. 7, 10303 (2016)). The interpretations and parameter dependences seem quite similar in the current work. Are there differences that could be attributed to the different materials or measurement techniques? I think a short statement about the “pro and contras” of BiSbTeSe_2 compared to other topological insulator-based Josephson junctions would be useful for the community.

  1. A hint why the results are similar is that the 4pi effect in these systems underlies a topological reason, as the Dirac states are immune to backscattering and that this “protection” is not sensitive to the material system but only the topological phase. Saying this, there are other signs that this effect seems, on the contrary, not so robust. The authors state that only two of over hundred junctions show 4Pi periodicity and that this could be an issue of sample qualities. However, I am surprised that a topological effect seems very fragile towards disorder (sample quality). It would be good if the authors could elaborate more on this issue.

  1. Actually, it is quite surprising that the authors see even a 4Pi effect, although that is consistent with other data. The experiment if I understand correctly is done in a sample that obeys time-reversal symmetry. The consequene of that is that the Andreev bound states are “touching” the continuum above the superconducting gap that is an unavoidable source of quasiparticle poisoning that would hinder the 4Pi effect. What is the effect of quasiparticle poisoning on the measurement and its interpretation? Is that the reason why the 4Pi effect is weak?

  1. Why the 4Pi effect is only seen at low frequencies of the microwave irradiation?

  1. In Fig. 1d) the Fraunhofer pattern is presented. The authors state that the Fourier transform (inset to the figure) “shows a moderately homogeneous current distribution through the junction”. Why does the Fourier transform show the spatial dependence of the current? Can the author explain that?

Some other points:

- In the caption of Fig. 1c), it is written “RT-curve..” although the plot shows the critical current I_c as a function of temperature T.

- Fig. 2 d) is the measurement at 9.6 GHz and the caption says: “The first Shapiro step is clearly absent”. I think it should the opposite, the first step is *not* absent.

The work is timely and should be relevant for the quantum transport community. The measurements seem solid and the interpretations conclusive.  I suggest the authors consider my points above to amend the presentation.

Author Response

We thank the Referee for the careful reading and the positive judgement of our manuscript and for the suggestions to further improve the manuscript. Here is our point-by-point reply:

  1. In our revised introduction (new paragraphs are indicated in blue), we now compare BiSbTeSe2 to other topological insulators and we argue why it is interesting to also study the RF response of Josephson junctions with this interlayer material. Indeed, the junction parameters are very similar to other materials systems for which the 4pi-periodicities have been reported. The key to success, in achieving the required ballistic transport, lies in the method of capping the topological crystals with h-BN.
  2. Indeed, the zero-mode Andreev bound state is a topological object, as the electrons in the bound state are protected from back-scattering. As BSTS has two-dimensional conducting surfaces, modes in other, non-perpendicular, directions are allowed, which are not protected from scattering (angles away from 180 degrees). This is the reason why disorder would still be detrimental to the topological modes. In 1D this would not be the case. We have added text and references to the introduction in order to clarify this point. Specifically, we have improved the sentence on the 100 samples. Namely, when we make a device with the h-BN top gate, then we observe the 4pi periodicity (100% success rate), whereas the >100 samples made in other ways did not reveal this (0% success). Likely, this has to do with the fact that the h-BN capped sample has less degradation induced defects, thereby allowing for ballistic transport.
  3. Correct, the quasiparticle poisoning is preventing us to see the 4pi-periodic states in dc measurements (e.g. in a SQUID). One has to measure faster than the quasiparticle poisoning rate, which is happening in our microwave experiment. We have added a sentence to the introduction about this.
  4. When both 2pi and 4pi modes are present in a Josephson junction, at higher frequencies the 2pi modes become dominantly visible when compared to the 4pi modes. This effect has been explained with the resistively shunted junction model by Wiedenmann et al. [15]. We have clarified this point in the revised text.
  5. The orbital contribution of the magnetic field to the vector potential gives a phase gradient along a Josephson junction. Together with the dc Josephson equation and after integrating along a closed contour this provides the famous Fraunhofer dependence of the critical current of a uniform junction as function of field. A full derivation of why the Ic(B) is related to the Fourier transform of the current distribution can, for example, be found in the text book: Barone and G. Paterno, Physics and Application of the Josephson Effect, Wiley, New York, 1982.

Additionally we have changed the captions of Fig. 1c and 2d according to the attentive comments of the Referee.

Round 2

Reviewer 2 Report

Second report.

It is strange that the authors do not want to make minor changes in the text which I have asked. While after replying the other referees the manuscript has become much clear, I any way insist on few additional changes, clarifying the paper. After these changes will be done, the paper can be accepted for publication.

0. I have a new comment after reading the resubmitted version with some clarifications. Since the authors have observed this interesting effect with h-BN protection layer, I ask them to add into the text few more details on how an h-BN flake was placed on top of the BiSbTeSe.

"1-3. Figure sizes and color schemes can, of course, be adjusted after consultation and upon request of the editor. It is of course a matter of taste but we are of the opinion that the preprint representation (figures with full page width) makes the figures large enough. Fig. 2a, in fact, does show the Shapiro steps. The fact that they are rounded is, in fact, a topic of the paper."

The Shapiro steps shown in Fig. 2a are almost invisible. That is why authors must recourse to binning to show them somehow. I am just asking to make the same plot for another signal power, where the Shapiro steps are better visible. This will allow readers to see if the steps are vertical enough and how much they are rounded. This is a common issue of researchers aiming to investigate physics, in comparison with others, designing sensitive cryogenic detectors. The first group of researchers does not pay much attention in suppression of EMI, appearing from compressors, while working with dry cryostats, and just make additional averaging, so masking the physical effects. And from Fig. 2a, since the Shapiro steps are too weak, one can not distinguish whether this is due to physics or due to low frequency noise. I believe that the authors have these data, and Fig. 2a corresponds to Fig. 2c for -9dBm signal. Then, if to re-plot Fig. 2a for -5dBm signal, the 1st step will be larger, in accordance with Fig. 2c. On the other hand, if they wish, the authors may plot both curves, the current one and the new one for -5dBm. As an example, perfectly vertical Shapiro steps can be seen again in IEEE Trans. Appl. Supercond., 28, 1100505 (2018) (or in many other similar papers).

"The measurement are done by standard current biasing, from negative to positive. The figure only presents the measured voltage. We mentioned the measurement configuration and we now write explicitly: “The junctions were characterized in a pseudo four-wire configuration with a combination of ac and dc current biasing.”"

This is OK now.

"We thank the Referee for the attentive reading. Indeed, this is a misprint, see also our reply to Referee 1. The caption of fig. 2d has been corrected to say, of course, that the first Shapiro step is present."

OK

"We consistently use the fact that the power dependence of the step does not go to zero, not the Fraunhofer magnetic field dependence which the Referee refers to. The paper that referee suggested does not not have any power dependence of the Shapiro steps, which makes it hard to compare to (only the non-oscillatory regime is relevant for the missing steps)."

Yes, I mean namely this. If at a certain power level the Fraunhofer magnetic field dependence does not go to zero, then it is a common feature that the power dependence will not also go to zero. If the authors now counter examples, let they tell them. If not, then the question arise, since the authors consider a rather new material, why they are sure that non-reaching a zero level for the power dependence of a Shapiro step is namely this effect which they claim? While the authors have partly clarified this question in new version of the manuscript, they may wish to add something else.

"The sine-Gordon equation only becomes necessary when the length (in our convention, the width W) is larger than the Josephson penetration length, $\lambda_J=\sqrt{\Phi_0/2\pi L I_c}$. Usually it is used in junctions with mA range critical current. At low temperatures, the inductance L is dominated by the kinetic inductance (Lk) in superconducting devices. We estimate the Lk in our experiment as Lk = (m/2 n_s e^2) l/A which is about 1 pH, and the critical current I_c is about 1 microAmp. This would safely put us in the short junction regime. "

This is namely the digits which I am asking to add into the manuscript. The authors should give their estimates in the text to show that for their samples it is not necessary to recourse to the sine-Gordon equation and their model is relevant.

Author Response

We thank the reviewer for the remaining suggestions. We apologize for not taking these properly into account in the first round (the reason being that this review only came available to us during the resubmission procedure of our manuscript based on the earlier 3 reports). Of course we are willing to take the advice into account. Please find below our point-by-point response.

0. We have included the following description of the fabrication steps concerning the h-BN capping flake: “We first prepared the PDMS/PMMA (1mm/A4-500 rpm) double layer on a piece of transparent glass, followed by a 10 min baking process at 70 \degree C. Then, the h-BN flakes were directly exfoliated onto the double layer. A suitable h-BN flake was identified under an optical microscope, after which it was put on top of the desired BSTS (1112) flake using a micro-manipulator. At the end, the entire PMMA layer was released by heating up the sample up to 130 \degree C.”

1-3. We now show the data for a power of -5dBm where the first step is again clearly visible. We fully agree that the steps are rounded by noise in these type of Josephson junctions with a critical current of just about a microampere. We have added a sentence to the manuscript concerning the noise rounding.

4-5. OK now

6. We are glad that we have already partly answered this question in the previous revisions. The fairly regular Fraunhofer dependence is show in Fig. 1d. Additionally, we have now added the following to the revised manuscript: “It is interesting to note that the lack of suppression holds for the first minimum (and subsequent odd minima) in the critical current versus applied microwave power, whereas the second minimum is very close to zero, just like the RSJ model predicts.” If there would have been a short or a pinhole, then one would not have expected this behavior.

7. We have now added the estimates of the short junction limit to the revised manuscript, as suggested.